



# LCM2021 – The UK Land Cover Map 2021

Christopher G. Marston[1], Aneurin W. O'Neil[1], R. Daniel Morton[1], Claire M. Wood[1], Clare S. Rowland[1]

[1]UK Centre for Ecology and Hydrology, Lancaster Environment Centre, Library Avenue, Bailrigg, Lancaster, LA1 4AP, UK

*Correspondence to*: Christopher G. Marston (cmarston@ceh.ac.uk)

**Abstract.** Land cover is a key environmental variable, underpinning widespread environmental research and decision-making. The UK Centre for Ecology and Hydrology (UKCEH) have provided reliable land cover information since the early 1990's; this supports multiple scientific, government and commercial objectives. Recent advances in computation and satellite data availability have enabled annual UKCEH land cover maps since 2017. Here we introduce the latest, annual UK Land Cover Map, representing 2021 (LCM2021) and describe its production and validation. LCM2021 methods replicate those for LCM2017 to LCM2020 with minor deviations to enhance accuracy. LCM2021 is based on the classification of satellite and spatial context data into 21 land cover/habitat classes, from which a product suite is derived. The production of LCM2021 involved three highly automated key stages: pre-processing of input data, image classification and production of the final data products. Google Earth Engine scripts were used to create an input data stack of satellite and context data. A set of training areas was created, based on data harvested from historic UKCEH land cover maps. The training data were used to construct a Random Forest classifier, which yielded classified images. Compiled results were validated against 35,182 reference samples, with correspondence tables indicating variable class accuracy and an overall accuracy of 82.6 % for the 21-class data and 86.5 % at a 10 aggregated-class level.

The UK Land Cover Map product suite includes a set of raster products in various projections, thematic and spatial resolutions (10 m, 25 m and 1 km) and land-parcel/vector product. The data are provided in 21-class (all configurations) and aggregated 10-class versions (1 km raster products only). All raster products are freely available for academic and non-commercial research. The data for Great Britain (GB) are provided in the British National Grid projection (https://epsg.io/27700) and the Northern Ireland (NI) data are in the TM75 Irish National Grid (https://epsg.io/29903). Information on how to access the data is given in the Data Availability section of the paper.

## 1 Introduction

Monitoring and managing environmental change is one of the key challenges for the 21st century (Turner et al., 2007; Allen et al., 2021). Land cover change is both a key cause, and consequence, of environmental change (Lambin et al., 2001; Foley et al., 2005), and as such it is recognised as a key variable for characterising the environment (Rockström et al., 2009; Bojinski et al., 2014). Land cover affects all aspects of the environment (Foley et al., 2005), including the hydrosphere (Teixeira et al., 2014), atmosphere (Allen et al., 2017) and biosphere (Oliver & Morecroft, 2014), as well as being able to compound or mitigate climate change (Morecroft et al., 2019). Land cover data are therefore an important starting point in



many environmental projects and analyses, as they form a basis against which other data sets may be integrated and
understood (e.g. Coxon et al., (2020)). Consequently, there is a demand nationally and internationally for accurate, timely
data on land cover. In the United Kingdom (UK), the demand for land cover data has been met by the UK Land Cover Map
(LCM) series, comprising LCMs for 1990, 2000, 2007, 2015, 2017, 2018, 2019, 2020 and now 2021. The UK LCMs are a
core part of the UK environmental data infrastructure providing data for a wide range of environmental applications and for
a diverse range of users, including academics, businesses, and government departments and agencies. Government use of
land cover data includes informing government decision-making by exploring the impact of different land-use scenarios
(Harrison et al., 2022), creating new data sets to aid implementation of conservation objectives (Natural England, 2022),
and providing for the UK's Natural Capital accounts (Office for National Statistics, 2021).

UK LCM data have proven valuable for commercial applications, typically in combination with other data and modelling,
to enable companies to better manage resources and target interventions. For example, water companies have used LCM
and modelling to optimise water quality monitoring in areas with high levels of agricultural run-off (United Utilities, 2017).
Additionally, telecommunications companies mapped locations of TV 'white space' (low/no signal) to target improvements
in poor signal areas using LCM and elevation data (Ishizu & Tran, 2014). LCM data has also enabled companies to make
better use of their land, with Highways England using LCM and data modelling to identify and remedy key gaps in
biodiversity corridors in their land holdings around roads across SW England (UKCEH, 2021). LCM has also been used in
data services for different sectors of UK industry, including underpinning flood modelling, where LCM data are used in the
Flood Estimation Handbook web service, the industry standard for assessing UK flood risk (FEH, 2018). The data have also
been used to conserve a protected species, by enabling the mapping of Great Crested Newt risk zones (Natural England,
2022) enabling a conservation partnership to sustainably manage the impact of development on newt populations (Tew &
Nicolet, 2019), and are increasingly used by environmental consultancies for estimating Natural Capital accounts (White et
al., 2015).

Academic uses of LCM data are wide-ranging, including applications in pollution, ecology, hydrology, meteorology and
climate change, with research topics motivated by both science and policy-related questions. Ecological applications have
included epidemiology (Gulliver et al., 2011), conservation (Hooftman & Bullock, 2012) and modelling spatial distributions
for mammals (Croft et al., (2017), insects (Mair et al., 2014), birds (Carrasco et al., 2018), invasive species (Fraser et al.,
2015) and pollination (Senapathi et al., *2015;* Baude et al., 2016). Whilst hydrological applications have included assessing
impacts of catchment land-use on rivers and lakes (Bussi et al., 2016), determining flood risk (Reynard et al., 2001; FEH,
2018) and modelling impacts of farming on water quality (Taylor et al., 2016). Spatial variability in health has also been
explored through modelling of hayfever risk (McInnes et al., 2017), air pollution impacts on human health (Stedman et al.,
1997) and bovine tuberculosis (Wint et al., 2002). In recent years, the LCM has also been used increasingly for mapping





ecosystem service provision (Emmett et al., 2016) and natural capital (Norton et al., 2018), and to aid creation of new data
sets such as the UKCEH Land Cover Plus: Pesticides 2012-2017 maps (Jarvis et al., 2020).

This paper describes the methods and data used to produce the UK Land Cover Map 2021 (LCM2021), as well as the derived
LCM2021 data products. LCM2021 was created by classifying satellite data into 21 land cover classes, with these classes
based on the UK Biodiversity Action Plan Broad Habitat definitions (Jackson, 2000). The LCM2021 production process
involved three stages: pre-processing of input data, image classification and production of the final data products. We present
the results of the classification and the validation of 21-class and 10-class versions of the data set. We describe the different
data products available and explain how they can be accessed.

## 2 Input data sets

Producing a Land Cover Map requires a range of data sets, typically including satellite data and context data, as well as
training and validation data. These data sets are described here, followed by the methods in Section 3.

### 2.1 Satellite data

LCM2021 used Sentinel-2 MultiSpectral Instrument (MSI) Level-2A surface reflectance satellite data (Drusch et al., 2012;
Claverie et al., 2018) acquired and pre-processed in Google Earth Engine (Gorelick et al., 2017). The images were acquired
between the 1st December 2020 and the 31st January 2022. All 10 and 20 m spectral bands, comprising bands 2 (490 nm), 3
(560 nm), 4 (665 nm), 5 (705 nm), 6 (740 nm), 7 (783 nm), 8 (842 nm), 8a (865 nm), 11 (1610 nm) and 12 (2190 nm) were
used.

### 2.2 Context data

Context data were used as additional inputs to the classification process to enable better classification of the required land
cover classes (Rogan et al., 2003). The context data included a digital elevation model (DEM), coastline, foreshore and tidal
water layers (to aid coastal classification), building and road layers (to reduce confusion between arable and urban areas)
(Table 1), and freshwater and forest layers. The DEM was used to calculate slope and aspect, which were also included as
context layers. National cartographic products for Great Britain (GB) were provided by the Ordnance Survey (OS), the
national mapping agency of GB, whereas for Northern Ireland (NI) products were provided by a number of government
organisations including the NI Statistics and Research Agency (NISRA), Ordnance Survey Northern Ireland (OSNI) and
the NI Department of Agriculture, Environment and Rural Affairs (DAERA). Slightly different context products were
available for NI compared to GB (Table 1). The main difference between the OS NI and OS GB context data, is the lack of
a NI equivalent to the GB buildings layer. The OS layers were converted from vector to raster data, with distance from
layers created for buildings, roads, rivers and water bodies. Distance from products were used to allow the context data





products to influence a wider area, rather than just the pixels they intersected with. The 10 m NEXTMap Digitial Elevation
Model (DEM) was used to calculate slope and aspect, with all three included as context layers.

**Table 1:** Context data set details, including comments on accessibility, data quality and timeliness. [1] Slope and aspect were
derived from the DEM data. Abbreviations: Great Britain (GB), Northern Ireland (NI). [2] data used subject to licensing
conditions, [3] data used under an open license. Ordnance Survey GB open data from: https://osdatahub.os.uk/, Ordnance
Survey NI data from: https://www.nidirect.gov.uk/articles/osni-open-data-product-list, NI Statistics and Research Agency
data from: https://www.opendatani.gov.uk/dataset/settlement-development-limits-2015, DAERA data sets from:
https://www.daera-ni.gov.uk/articles/wmu-digital-dataset-downloads, Copernicus Land Monitoring Service data sets from:
https://land.copernicus.eu/pan-european/corine-land-cover/clc-2012.

| Type of data set | Rationale | Extent | Data provider: | Data set name |
|---|---|---|---|---|
| **Topographical** | Constrain land cover classes to appropriate slopes and altitudes. | GB | Nextmap [2] | Digital elevation data[1] |
| | | NI | Ordnance Survey Northern Ireland (OSNI) [3] | 10m Digital Terrain Model height data |
| **Urban extent** | Distance from urban and roads, used to limit spectral confusion, especially between arable and urban. | GB | Ordnance Survey (OS) [3]  Copernicus Land Monitoring Service [3] | OS Vectormap District, building polygons; OS Open roads  Corine Land Cover 2012, airport polygons |
| | | NI | OSNI [3]  NI Statistics and Research Agency [3]  Copernicus Land Monitoring Service [3] | Open Data 50k Transport Lines; Settlement development limits  Corine Land Cover 2012, airport polygons |
| **Coastal** | Constrain coastal classes so they do not appear inland. Coastal context layer include foreshore extent, tidal water extent and distance to mean high water line. | GB | OS [3] | OS Terrain 50 |
| | | NI | Department of Agriculture, Environment and Rural Affairs (DAERA) [3] | Marine Digital Datasets |
| **Water** | Distance from water used to improve classification of | GB | OS [3] | Open Map Local, surface water area polygons |





| | | | | |
|---|---|---|---|---|
| | habitats often associated with proximity to rivers (e.g. Fen, Marsh and Swamp, and Neutral Grassland). | NI | DAERA [3] | Rivers Digital Datasets – River segments; Lakes Digital Datasets – Lake water bodies. |
| **Forest** | Improve extent of forest, especially for recently harvested forest and newly planted forest. | GB | OS [3] | OS Vectormap District woodland polygons |


## 2.3 Training area data

LCM2021 is produced through supervised classification of satellite images, an empirical process that requires training areas of known land cover type. The training areas for the classification were widely distributed to capture the range of spectral signatures typical of each class. For LCM2021 training areas were primarily harvested from existing vector data from LCM2018 (Morton et al., 2020a, b), LCM2019 (Morton et al., 2020c, d) and LCM2020 (Morton et al., 2021e, f). The method is described in section 3.2.1.

## 2.4 Spatial framework

The LCM spatial framework is a set of land parcel polygons summarising the landscape of the UK into real world objects such as lakes, fields, woodlands and urban sites. It was derived from generalised digital cartography (Ordnance Survey MasterMap[TM] topographic layer (OSMM) for GB and Ordnance Survey of Northern Ireland (OSNI) Large-scale Vector for NI), supplemented with rural payment boundary data (Smith et al., 2007; Morton et al., *2011,*). The spatial framework was first generated for LCM2007 and revised for LCM2015 onwards, by fixing some minor spatial errors and additional simplifications of land parcel structure. The spatial framework is used to derive a land parcel dataset from which 25 m and 1 km raster datasets are generated.

## 2.5 Validation data sets

Validation data are necessary to establish the accuracy of land cover classifications (Foody et al., 2002). LCM2021 validation used a UK-wide data set of 35,182 points gathered from field observations, manual interpretation of aerial photography and quality assured third party data sets (Fig. 1). The validation data, included habitat mapping and plot data from Countryside Survey data (Wood et al., 2017), supplemented with additional points for arable land (8589 points) collected in 2020 by the Rural Payments Agency. Data from the National Forest Inventory (NFI, 2019) was used to validate the broadleaved woodland and coniferous woodland classes for GB. Further data were gathered from the 2007 LCM validation field survey (Morton et al., 2011a) checked against current (circa-2021) aerial photography to ensure no change



had occurred, some additional manually derived points (interpreted from aerial photography) were also added, particularly
for water and urban-classes.

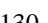


**Figure 1: Distribution of the 35,182 validation points for LCM2021.**



## 3 Methods

Figure 2 shows the key stages in the creation of LCM2021, from image acquisition through to the creation of the final suite of data products.

**Pre-processing**

- Select images
- Cloud mask images
- Create cloud-free composite
- Combine composite and ancillary data

**Classification**

- Training data
- Classify
- Iterate
- Review
- Finished classification

**Final data production**

- Mosaic finished classifications
- Ingest into spatial framework
- Create final products

**Figure 2: Overview of the processing workflow, showing the three main production phases.**

### 3.1 Composite image creation

Temporal composite images (also known as temporal aggregations) are increasingly used to compress voluminous image collections and overcome problems of data gaps caused by clouds in optical imagery (Carrasco et al., 2019; Holben, 1986). Cloud computation platforms, such as Google Earth Engine (Gorelick et al., 2017), provide users with tools to create composite images aggregated over user-defined intervals (e.g. annually, monthly, bi-monthly) and for user-defined properties (e.g. raw bands, spectral indices) and with user-defined functions (e.g. median, maximum, mean).

Seasonal composites images of Sentinel-2 Multi-Spectral Instrument (MSI) Level 2-A data (Drusch et al., 2012) were created using Google Earth Engine, with cloud-masking based on the Sentinel-2 Cloud Probability layer, s2cloudless (Skakun et al., 2022) and snow-masking based on the QA attributes performed. Images representing median surface reflectance were aggregated over four composite periods; 1st Dec 2020 - 31st March 2021; 1st April 2021 – 30th June 2021;





1st July 2021 – 31st September 2021; and 1st October 2021 – 31st Jan 2022. Periods one and four were extended by a month
into the previous and following years to reduce cloud effects. Seasonal composites were used as they capture the variability
in vegetation phenology through the year, which aids separation of the different land cover classes (Carrasco et al., 2019),
and for the UK, aggregation of Sentinel-2 data for four seasons provides data for all four seasons in over 99.9% of pixels
(see SI). Context layers including slope, aspect, elevation distance to coast, distance to building, distance to road, distance
to freshwater, plus a foreshore mask, tidal water mask and a forest mask (GB only) were integrated with the composite
period satellite imagery (see Table 1 for details of the context data layers, which varied slightly between GB and NI). The
addition of context layers reduces spectral confusion between different classes with similar spectral characteristics. The
seasonal composites, with the added context data, were then classified.
**3.2 Classification**
LCM2021 is based on the 21-class nomenclature presented in Table 2. The 21 Land Cover classes are based on UK Broad
Habitat definitions (Jackson, 2000) and are designed to cover the range of habitats found in the UK that can be reliably
mapped from satellites. Detailed descriptions of the classes are given in Appendix 1. Production of the classifications is split
into two stages, first developing the core training areas (section 3.2.1), and second the classification process (section 3.2.2).

**Table 2. Relationship between the 21 LCM2021 classes, the 10 Aggregate classes and the underlying Broad Habitat classes. Italic**
**text highlights classes meeting the Broad Habitats classes as documented in Jackson (2000). [1,2] LCM2021 and Aggregate class**
**numbers are used for raster data sets.**

| LCM2021 Aggregate class | LCM2021 aggregate class number[1] | LCM2021 Target class | LCM2021 target class number[2] | Associated Broad Habitat |
|---|---|---|---|---|
| Broadleaf woodland | 1 | Broadleaved woodland | 1 | *'Broadleaved, mixed and yew woodland'* |
| Coniferous woodland | 2 | *'Coniferous woodland'* | 2 | *'Coniferous woodland'* |
| Arable | 3 | *'Arable and horticulture'* | 3 | *'Arable and horticulture'* |
| Improved grassland | 4 | *'Improved grassland'* | 4 | *'Improved grassland'* |
| Semi-natural grassland | 5 | *'Neutral grassland'* | 5 | *'Neutral grassland'* |
| | | *'Calcareous grassland'* | 6 | *'Calcareous grassland'* |
| | | Acid grassland | 7 | *'Acid grassland'* |
| | | *'Fen, marsh and swamp'* | 8 | *'Fen, marsh and swamp'* |
| Mountain, heath, bog | 6 | Heather | 9 | *'Dwarf shrub heath'* |
| | | Heather grassland | 10 | |
| | | *'Bog'* | 11 | *'Bog'* |
| | | *'Inland rock'* | 12 | *'Inland rock'* |
| Saltwater | 7 | Saltwater | 13 | Saltwater |



| Freshwater | 8 | Freshwater | 14 | Freshwater | |
| Coastal | 9 | *'Supra-littoral rock'* | 15 | *'Supra-littoral rock'* | |
| | | *'Supra-littoral sediment'* | 16 | *'Supra-littoral sediment'* | |
| | | *'Littoral rock'* | 17 | *'Littoral rock'* | |
| | | Littoral sediment | 18 | *'Littoral sediment'* | |
| | | Saltmarsh | 19 | | |
| Built-up areas and gardens | 10 | Urban | 20 | *'Built-up areas and gardens'* | |
| | | Suburban | 21 | | |



### 3.2.1 Core training areas

Selecting appropriate training areas is crucial for accurate classification of satellite data and has traditionally been time-consuming. LCM2021 used a method based on training areas that remained stable across the three previous maps (LCM2018, LCM2019 and LCM2020) on the assumption that many areas such as woodland and urban areas remain stable over decades. Identifying such areas provides a core data set as a starting point for each classification, with this core dataset undergoing edits where required to produce the final classification.

When selecting training polygons from this spatial framework, as well as identifying polygons classified as the same land cover class for LCM2018/19/20, these polygons were also required to have a purity value of >80% in each of the three land cover classifications to be included. The purity value of a polygon is a measure of the percentage of the modal land cover class, over the total number of pixels corresponding to that polygon. The 80% threshold was selected to retain a high level of purity within the training polygons, but to retain a large enough set of polygons within each classification extent, with the aim of achieving a spatially distributed training data set with a good representation of all land cover classes. Some incorrect training polygons were present within this core training data set, due to either misclassifications in the earlier Land Cover Maps, or because of changes in land cover. Systematic visual checks of the training data and the resultant classifications aided in identifying and removing inappropriate polygons.

### 3.2.2 Classification algorithm

The composite images were classified using the Random Forest algorithm (Breiman, 2001) in the WEKA package (Hall et al., 2009; Frank et al., 2016). For each of the tiles, a Random Forest classifier based on 200 trees was trained. When building a Random Forest classifier it is important to balance the training samples. An unbalanced classifier will bias towards common classes and rare classes may be lost from results completely. Balance was achieved by bagging all training pixels



per class, then from each bag sampling 10000 pixels with replacement. For each pixel the balanced RF classifier yields a
probability of membership for all 21 land cover classes. Land cover per pixel is assigned by highest probability.

### 3.3 Product construction

Classifications for all tiles were compiled into a full UK spatial coverage at 10 m pixel resolution. This produced a two-
band image. Band one is the most likely land cover; band two the probability associated with this land cover, but rescaled
into a integer over the interval 0 to 100. Rescaling to an integer enables classification results to be stored in 8-bit, thereby
reducing data size without degrading information. The 10 m raster is the precursor for all derived products.

The ingestion into the spatial framework involved determining the majority (modal) class for each polygon. Separate GB
and NI data sets were created to accommodate the different map projections. Figure 3 shows the extents of the 32 composites
used to achieve complete coverage of the UK. The approximate 100x100 km tile size, based on a modified version of the
Ordnance Survey 100 km tile grid was chosen as this provides a manageable size for processing. Some tiles such as those
encompassing the Western Isles, Orkney and Shetland, and Cornwall and the Scilly Isles are intentionally enlarged to avoid
a sparsity of training data due to the extensive presence of sea in these tiles. Occasionally where tile extents are modified,
overlap between adjacent tiles does occur.





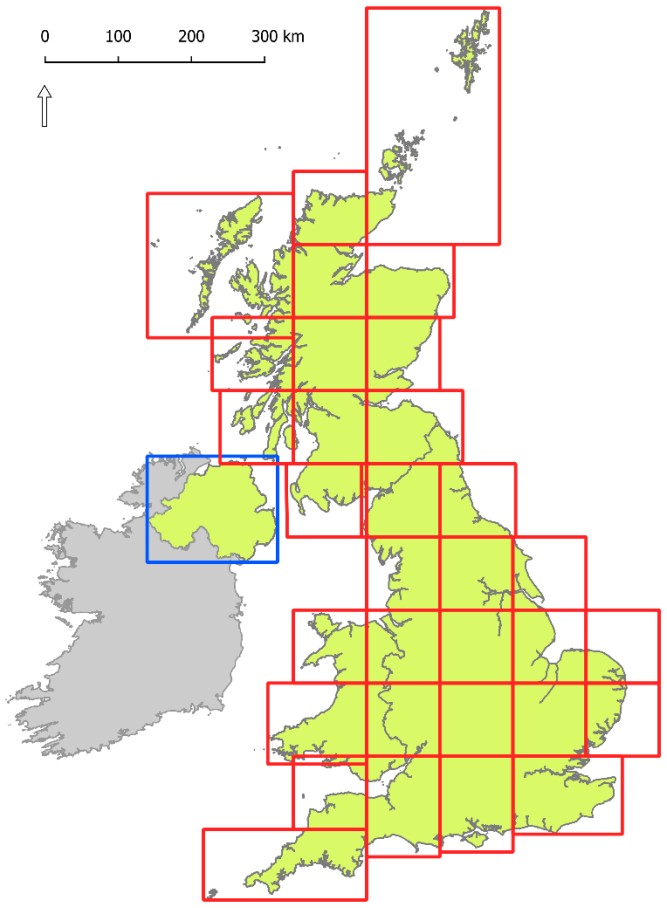


**Figure 3: Composite images extents comprising LCM2021 for Great Britain (red) and Northern Ireland (blue).**


Once the GB and NI classification mosaics were complete, a series of minor knowledge-based corrections were applied.
These included reclassification of misclassified arable pixels to improved grassland in urban green space areas (as denoted
by the OS Open Greenspace data set), and of coastal classes being misclassified inland using a coastal mask.

### 3.4 Validation

The LCM2021 class was extracted for each of the validation points. From this data, confusion matrices were plotted for the
21 target classes and the 10 aggregate classes used for LCM2021.



# 4 Results

## 4.1 Validation results

The 25 m rasterised polygon version of LCM2021 (Marston et al., 2022e, f) was validated using 35,182 points distributed across the UK (Table 3). The results are summarised in a confusion matrix, which shows how reference points for each of the classes were classified. Ideally, all the points would fall along the main diagonal (highlighted in green in Table 3), showing complete agreement between the reference data and the classification. Table 3 shows that LCM2021 has an overall accuracy of 82.6%, with the accuracy of individual classes varying. The results of the validation are shown in a confusion matrix (Table 3), with the reference data in the columns and the classification data in the rows. The confusion matrix shows the level of agreement between the classification and the reference data, as well as the areas of disagreement or confusion. The accuracy varies with class, with the Producer's Accuracy ranging between high and low values of 93.9% (saltmarsh) and 35.4% (heather grassland), and the User's Accuracy varying between 96.1% (arable) and 42.6% (heather grassland). For the products that use the 10 aggregate classes (see section 5 for more details about the aggregate class products) the validation suggests an overall accuracy of 86.5% (Table 4).

**Table 3. Correspondence matrix for LCM2021 against 35,182 reference points. The main diagonal is highlighted green. BW = Broadleaved woodland; CW = Coniferous woodland; AR = Arable; IG = Improved grassland; NG = Neutral grassland; CG = Calcareous grassland; AG = Acid grassland; FMS = Fen, marsh, swamp; H = Heather; HG = Heather grassland; B = Bog; IR = Inland rock; SW = Saltwater; FW = Freshwater; SLR = Supra-littoral rock; SLS = Supra-littoral sediment; LR = Littoral rock; LS = Littoral sediment; SM = Saltmarsh; U = Urban; SU = Suburban; PA = Producer's accuracy; UA = User's accuracy; OA = Overall accuracy.**



| Classified Data | Reference Data | | | | | | | | | | | | | | | | | | | | | Total | UA (%) |
|---|---|---|---|---|---|---|---|---|---|---|---|---|---|---|---|---|---|---|---|---|---|---|---|
| | BW | CW | AR | IG | NG | CG | AG | FMS | H | HG | B | IR | SW | FW | SLR | SLS | LR | LS | SM | U | SU | | |
| BW | 1704 | 218 | 19 | 73 | 24 | 0 | 10 | 0 | 5 | 26 | 2 | 4 | 0 | 9 | 1 | 0 | 0 | 1 | 1 | 2 | 13 | 2112 | 80.7 |
| CW | 55 | 649 | 3 | 1 | 1 | 1 | 4 | 0 | 5 | 16 | 3 | 0 | 0 | 0 | 0 | 0 | 0 | 0 | 0 | 1 | 0 | 739 | 87.8 |
| AR | 22 | 3 | 10102 | 306 | 28 | 1 | 2 | 0 | 0 | 3 | 1 | 10 | 0 | 6 | 0 | 0 | 0 | 0 | 1 | 13 | 13 | 10511 | 96.1 |
| IG | 100 | 4 | 1027 | 4835 | 186 | 55 | 175 | 18 | 1 | 82 | 29 | 4 | 0 | 8 | 0 | 26 | 1 | 0 | 4 | 9 | 24 | 6588 | 73.4 |
| NG | 18 | 11 | 39 | 230 | 503 | 0 | 13 | 19 | 2 | 4 | 0 | 0 | 0 | 3 | 0 | 0 | 0 | 1 | 0 | 2 | 0 | 845 | 59.5 |
| CG | 30 | 4 | 26 | 47 | 4 | 946 | 15 | 0 | 1 | 6 | 0 | 13 | 0 | 0 | 0 | 0 | 0 | 0 | 0 | 1 | 0 | 1093 | 86.6 |
| AG | 20 | 1 | 88 | 177 | 6 | 55 | 1245 | 0 | 29 | 228 | 39 | 1 | 0 | 1 | 0 | 0 | 0 | 0 | 0 | 0 | 4 | 1894 | 65.7 |
| FMS | 15 | 0 | 5 | 14 | 2 | 1 | 4 | 577 | 0 | 0 | 2 | 0 | 0 | 8 | 0 | 0 | 0 | 0 | 0 | 0 | 0 | 628 | 91.9 |
| H | 7 | 1 | 9 | 1 | 0 | 0 | 30 | 1 | 819 | 104 | 121 | 0 | 0 | 0 | 1 | 0 | 0 | 0 | 0 | 0 | 0 | 1094 | 74.9 |
| HG | 17 | 3 | 9 | 12 | 0 | 5 | 158 | 3 | 81 | 299 | 106 | 4 | 0 | 3 | 0 | 1 | 0 | 0 | 0 | 0 | 1 | 702 | 42.6 |
| B | 0 | 3 | 0 | 3 | 0 | 3 | 31 | 4 | 27 | 71 | 877 | 0 | 0 | 1 | 0 | 0 | 0 | 0 | 0 | 0 | 0 | 1020 | 86.0 |
| IR | 0 | 0 | 10 | 2 | 1 | 9 | 4 | 0 | 3 | 2 | 2 | 125 | 0 | 0 | 0 | 0 | 0 | 0 | 0 | 8 | 0 | 166 | 75.3 |
| SW | 0 | 0 | 0 | 0 | 0 | 0 | 0 | 0 | 0 | 0 | 0 | 0 | 73 | 0 | 0 | 0 | 0 | 13 | 0 | 0 | 0 | 86 | 84.9 |
| FW | 13 | 0 | 1 | 4 | 1 | 0 | 1 | 2 | 0 | 0 | 0 | 0 | 0 | 548 | 0 | 0 | 0 | 0 | 0 | 0 | 2 | 572 | 95.8 |
| SLR | 0 | 0 | 0 | 5 | 0 | 0 | 7 | 0 | 0 | 0 | 0 | 0 | 0 | 0 | 42 | 6 | 11 | 6 | 0 | 0 | 0 | 77 | 54.5 |
| SLS | 1 | 0 | 3 | 2 | 4 | 0 | 1 | 0 | 0 | 0 | 0 | 0 | 0 | 0 | 1 | 178 | 0 | 7 | 1 | 0 | 0 | 198 | 89.9 |
| LR | 0 | 0 | 0 | 0 | 0 | 0 | 0 | 0 | 0 | 1 | 0 | 1 | 0 | 0 | 16 | 5 | 86 | 17 | 0 | 1 | 0 | 127 | 67.7 |
| LS | 0 | 0 | 0 | 0 | 0 | 0 | 0 | 0 | 0 | 0 | 0 | 0 | 9 | 0 | 0 | 11 | 7 | 211 | 3 | 0 | 0 | 241 | 87.6 |
| SM | 2 | 0 | 0 | 5 | 1 | 0 | 0 | 13 | 0 | 1 | 0 | 0 | 0 | 2 | 1 | 8 | 1 | 9 | 169 | 0 | 0 | 212 | 79.7 |
| U | 19 | 0 | 12 | 29 | 15 | 0 | 1 | 1 | 0 | 0 | 0 | 26 | 0 | 3 | 0 | 7 | 1 | 2 | 1 | 2343 | 303 | 2763 | 84.8 |
| SU | 151 | 1 | 17 | 223 | 44 | 0 | 2 | 0 | 1 | 1 | 2 | 10 | 0 | 3 | 0 | 3 | 0 | 0 | 0 | 329 | 2727 | 3514 | 77.6 |
| Total | 2174 | 898 | 11370 | 5969 | 820 | 1076 | 1703 | 638 | 974 | 844 | 1184 | 198 | 82 | 595 | 62 | 245 | 107 | 267 | 180 | 2709 | 3087 | 35182 | |
| PA (%) | 78.4 | 72.3 | 88.8 | 81.0 | 61.3 | 87.9 | 73.1 | 90.4 | 84.1 | 35.4 | 74.1 | 63.1 | 89.0 | 92.1 | 67.7 | 72.7 | 80.4 | 79.0 | 93.9 | 86.5 | 88.3 | | OA (%) = 82.6 |

Table 4: Correspondence matrix for LCM2021 aggregate classes against 35,182 reference points. The main diagonal is highlighted green. BW = Broadleaved woodland; CW = Coniferous woodland; AR = Arable; IG = Improved grassland; SNG = Semi-natural grassland; MHB = Mountain, heath and bog; SW = Saltwater; FW = Freshwater; C = Coastal; BU = Built-up and gardens; PA = Producer's accuracy; UA = User's accuracy; OA = Overall accuracy.

| Classified Data | Reference Data | | | | | | | | | | Total | UA (%) |
|---|---|---|---|---|---|---|---|---|---|---|---|---|
| | BW | CW | AR | IG | SNG | MHB | SW | FW | C | BU | | |
| BW | 1704 | 218 | 19 | 73 | 34 | 37 | 0 | 9 | 3 | 15 | 2112 | 80.7 |
| CW | 55 | 649 | 3 | 1 | 6 | 24 | 0 | 0 | 0 | 1 | 739 | 87.8 |
| AR | 22 | 3 | 10102 | 306 | 31 | 14 | 0 | 6 | 1 | 26 | 10511 | 96.1 |
| IG | 100 | 4 | 1027 | 4835 | 434 | 116 | 0 | 8 | 31 | 33 | 6588 | 73.4 |
| SNG | 83 | 16 | 158 | 468 | 3390 | 325 | 0 | 12 | 1 | 7 | 4460 | 76.0 |
| MHB | 24 | 7 | 28 | 18 | 249 | 2641 | 0 | 4 | 2 | 9 | 2982 | 88.6 |
| SW | 0 | 0 | 0 | 0 | 0 | 0 | 73 | 0 | 13 | 0 | 86 | 84.9 |



| | | | | | | | | | | | | |
|---|---|---|---|---|---|---|---|---|---|---|---|---|
| **FW** | 13 | 0 | 1 | 4 | 4 | 0 | 0 | 548 | 0 | 2 | 572 | **95.8** |
| **C** | 3 | 0 | 3 | 12 | 26 | 3 | 9 | 2 | 796 | 1 | 855 | **93.1** |
| **BU** | 170 | 1 | 29 | 252 | 63 | 40 | 0 | 6 | 14 | 5702 | 6277 | **90.8** |
| **Total** | 2174 | 898 | 11370 | 5969 | 4237 | 3200 | 82 | 595 | 861 | 5796 | 35182 | |
| **PA (%)** | **78.4** | **72.3** | **88.8** | **81.0** | **80.0** | **82.5** | **89.0** | **92.1** | **92.5** | **98.4** | OA: 86.5 % | |


## 4.2 LCM2021 map

The final LCM2021 product shows the expected distribution of classes across the UK (Fig. 4). At the scale shown in Fig. 4
the differences between the grassland of the west, and the arable areas in the east are clear, as are the uplands in Wales and
Scotland, with London, the UK's largest urban area, clearly visible.



Earth System
Science
Data

**Legend:**

- Broadleaved woodland
- Coniferous woodland
- Arable
- Improved grassland
- Neutral grassland
- Calcareous grassland
- Acid grassland
- Fen, marsh, swamp
- Heather
- Heather grassland
- Bog
- Inland rock
- Saltwater
- Freshwater
- Supra-littoral rock
- Supra-littoral sediment
- Littoral rock
- Littoral sediment
- Saltmarsh
- Urban
- Suburban

0    100    200    300 km

251

**Figure 4: LCM2021 in standard colour palette (see Table B2 for palette details) (see Appendix B for LCM2021 in revised colour palette).**



## 4.3 LCM statistics

One of the uses of LCM2021 is to produce country level statistics (Table 5), although land cover statistics can also be produced for other types of spatial units, such as river or lake catchments, or national parks and protected areas.

**Table 5: UK Land Cover Statistics derived from LCM2021 in area (km$^2$) calculated from the 10 m raster product.**

| Land Cover Code | Land cover class | UK | England | Scotland | Wales | Northern Ireland |
|---|---|---|---|---|---|---|
| 1 | Broadleaved woodland | 21045 | 12322 | 5330 | 2555 | 838 |
| 2 | Coniferous woodland | 13830 | 2788 | 9022 | 1422 | 598 |
| 3 | Arable | 49121 | 41867 | 5960 | 841 | 453 |
| 4 | Improved grassland | 66394 | 39304 | 13053 | 7765 | 6272 |
| 5 | Neutral grassland | 4200 | 1659 | 105 | 525 | 1911 |
| 6 | Calcareous grassland | 2561 | 2387 | 31 | 11 | 132 |
| 7 | Acid grassland | 21873 | 4448 | 12281 | 4404 | 740 |
| 8 | Fen | 783 | 471 | 68 | 182 | 62 |
| 9 | Heather | 11562 | 2081 | 8636 | 566 | 279 |
| 10 | Heather grassland | 11842 | 1433 | 9719 | 409 | 281 |
| 11 | Bog | 10457 | 1986 | 7255 | 251 | 965 |
| 12 | Inland rock | 2685 | 245 | 2362 | 63 | 15 |
| 13 | Saltwater | 935 | 720 | 53 | 4 | 158 |
| 14 | Freshwater | 3267 | 1093 | 1499 | 96 | 579 |
| 15 | Supra-littoral rock | 390 | 62 | 252 | 66 | 10 |
| 16 | Supra-littoral sediment | 723 | 169 | 340 | 102 | 112 |
| 17 | Littoral rock | 432 | 84 | 340 | 1 | 7 |
| 18 | Littoral sediment | 1444 | 1248 | 78 | 34 | 84 |
| 19 | Saltmarsh | 923 | 552 | 272 | 95 | 4 |
| 20 | Urban | 4901 | 4066 | 482 | 227 | 126 |
| 21 | Suburban | 17539 | 13669 | 1812 | 1308 | 750 |
| | Total area (km$^2$) | 246902 | 132651 | 78949 | 20927 | 14375 |



## 5 LCM2021 data products

LCM2021 is provided in a range of open data formats and at a range of thematic and spatial resolutions to support the needs of a wide range of users and applications. There are 21 target classes in the full thematic resolution product and 10 classes in the aggregated products (Table 2). The 'base' product is the 10 m raster (Marston et al., 2021a, b) from which all other products are derived (Fig. 5). The LCM2021 10 m raster is ingested into the spatial framework to produce a vector version of the data set (Marston et al., 2021 c, d). The vector version of the data set is then used to create a rasterised polygon version of the data set with a 25 m pixel size (Marston et al., 2022 e, f). The 25 m version is effectively the 'legacy' style land cover map and maintains a spatial consistency with the earlier Landsat-based Land Cover Maps of LCM1990 (Rowland et al., 2020a, b), LCM2007 (Morton et al., 2011b; Morton et al., 2014) and LCM2015 (Rowland et al., 2017a, b); LCM2000 (Fuller et al., 2002 a, b) currently uses a different spatial structure. The 25 m raster product is then used to produce the 1 km percentage cover and dominant cover products for both the 21 target classes and the 10 aggregate classes (Marston et al., 2022g). The Great Britain and Northern Ireland data sets are provided separately, with the GB data in British National Grid projection (EPSG:27700) and the Northern Ireland data in the Irish National Grid projection (EPSG:29903).

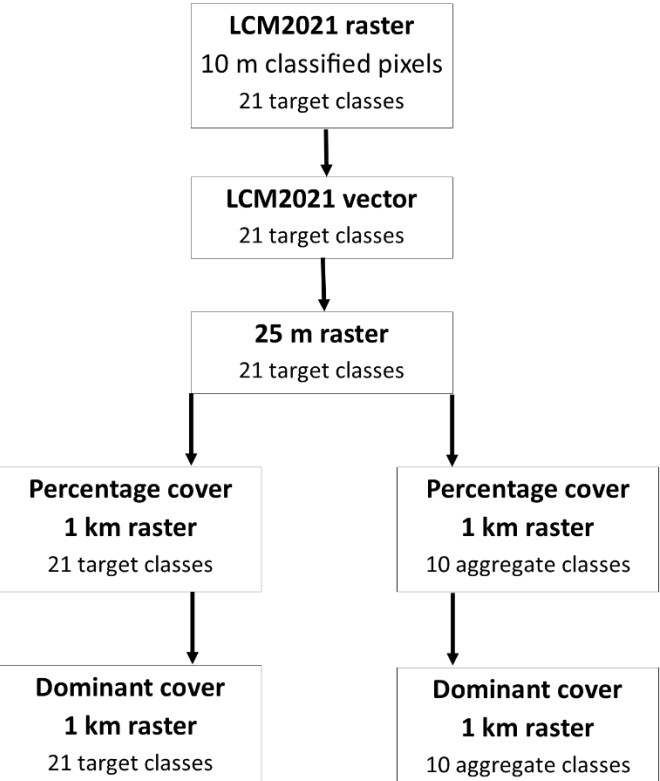

**Figure 5: Overview of the LCM2021 data set production process. The vector version of the data set is constrained by licensing restrictions due to the inclusion of national mapping agency data**


LCM2021 is produced in a range of spatial resolutions (Fig. 6) to support different types of analysis. The 10 m data set is a
relatively new data set (first produced in LCM2020) and enabled by the 10 m resolution of the optical Sentinel-2 bands. The
higher spatial resolution products capture the fine detail of the landscape and are often used for assessment of landscape
features requiring fine resolution, such as habitat connectivity (Hooftman & Bullock, 2012) or for detailed studies of small
areas (e.g. Miller et al., 2020). The 1 km data sets are primarily used for national-scale modelling, often in conjunction with
a range of other coarser resolution environmental data sets (e.g. Coxon et al., 2020; Jordan et al., 2022) and are useful for
showing the distribution of a particular class across the UK. For example, Fig. 7 shows the distribution of the broadleaf
woodland class and the urban class from the aggregated 1 km percentage data sets for the UK.

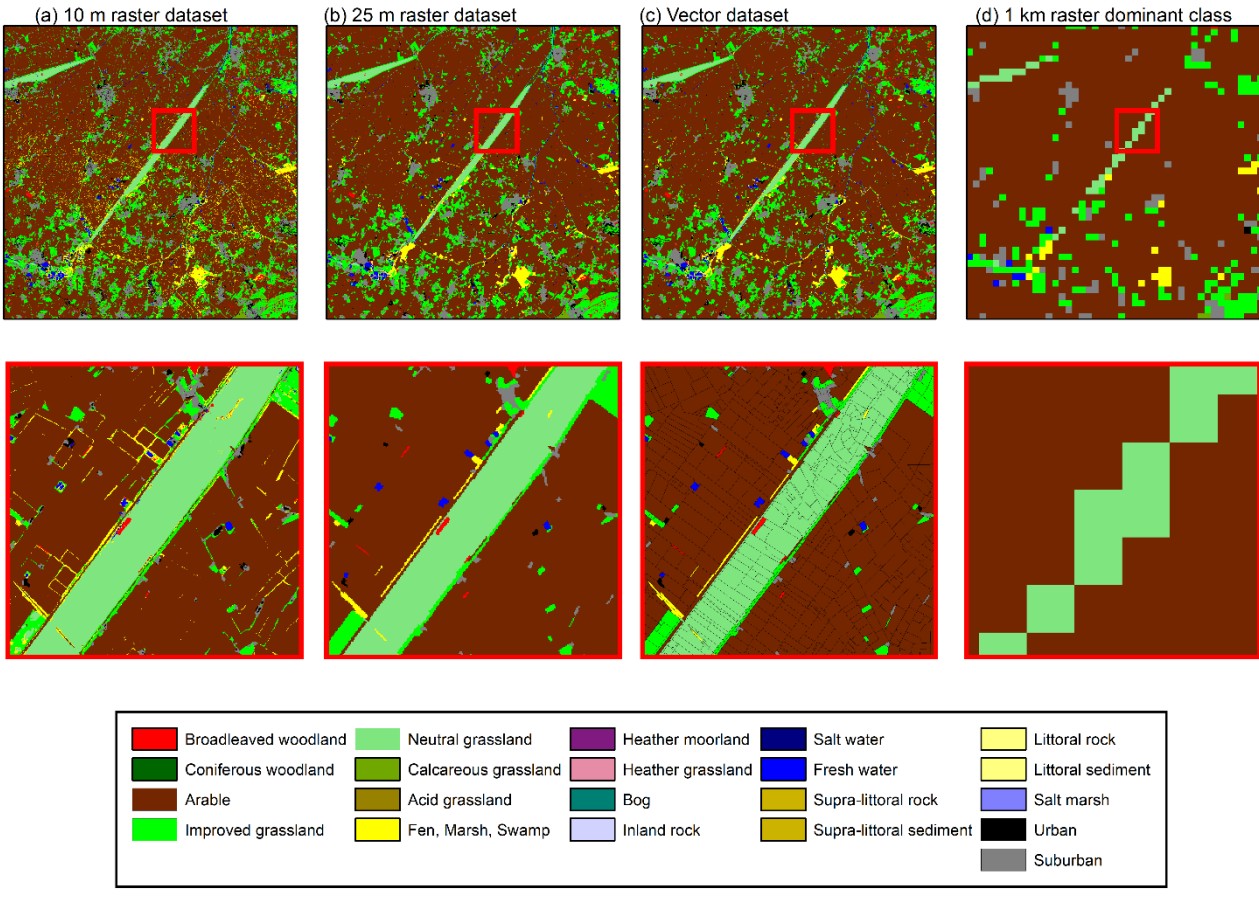


**Figure 6: Examples of the level of spatial detail provided by the (a) 10 m raster; (b) 25 m rasterised polygons; (c) vector data set,**
**and; (d) 1 km raster data sets. Top panels shows zoomed out view, red box shows the location of the zoomed in area in the lower**
**panels.**


**Figure 7: Examples of the UK-wide distribution of (a) Broadleaf woodland and (b) Built-up areas and gardens percentage cover, based on the 1 km aggregate class percentage data sets for GB and NI (Marston et al., 2021g).**

## 6 Data availability

The LCM2021 data products (Table 6) have digital object identifiers (doi) and are available via the NERC Environmental Data Service (https://eds.ukri.org/environmental-data-service), with all versions listed on the LCM2021 data collection page (UKCEH, 2022a). Raster data are provided as uncompressed GeoTiffs and are supplied with data set documentation, and QGIS files for displaying the classifications in the LCM-standard palette (used since LCM2000) (see Appendix B, B1 for example) and a palette designed to aid users affected by colour-vision deficiency (see Fig. 1 for example). The 10 m raster data sets are also viewable via a Web Mapping Service (UKCEH, 2022b).



Table 6: Digital Object Identifier (DOI) for the LCM2021 openly available products.

| Product | Region | DOI | Reference |
|---|---|---|---|
| **10 m classified pixels** | GB | https://doi.org/10.5285/a22baa7c-5809-4a02-87e0-3cf87d4e223a | Marston et al. (2022a) |
| | NI | https://doi.org/10.5285/e44ae9bd-fa32-4aab-9524-fbb11d34a20a | Marston et al. (2022b) |
| **25 m rasterised land parcels** | GB | https://doi.org/10.5285/a1f85307-cad7-4e32-a445-84410efdfa70 | Marston et al. (2022e) |
| | NI | https://doi.org/10.5285/f3310fe1-a6ea-4cdd-b9f6-f7fc66e4652e | Marston et al. (2022f) |
| **1 km summary raster data** | GB and NI | https://doi.org/10.5285/a3ff9411-3a7a-47e1-9b3e-79f21648237d | Marston et al. (2022g) |


The LCM2021 data products (Table 6) have digital object identifiers (DOI) and are available via the NERC Environmental
Data Service (https://eds.ukri.org/environmental-data-service), with all versions listed on the LCM2021 data collection page
(UKCEH, 2022a). Raster data are provided as uncompressed GeoTiffs and are supplied with data set documentation, and
QGIS files for displaying the classifications in the LCM-standard palette (used since LCM2000) (see Appendix B, B1 for
example) and a palette designed to aid users affected by colour-vision deficiency (see Fig. 1 for example). The 10 m raster
data sets are also viewable via a Web Mapping Service
To download the LCM2021 datasets for review purposes, anonymous data access is possible using the login credentials
username 'reviewer@eidc.ac.uk' and password 'reviewlcm2021'. This login (valid for 3 months) enables data access using
the following links: Great Britain 10 m classified pixels: https://catalogue.ceh.ac.uk/datastore/eidchub/a22baa7c-5809-
4a02-87e0-3cf87d4e223a/gblcm10m2021.tif; Northern Ireland 10 m classified pixels:
https://catalogue.ceh.ac.uk/datastore/eidchub/e44ae9bd-fa32-4aab-9524-fbb11d34a20a/nilcm10m2021.tif; Great Britain 25
m rasterised land parcels: https://catalogue.ceh.ac.uk/datastore/eidchub/a1f85307-cad7-4e32-a445-
84410efdfa70/gblcm25m2021.tif; Northern Ireland 25 m rasterised land parcels:
https://catalogue.ceh.ac.uk/datastore/eidchub/f3310fe1-a6ea-4cdd-b9f6-f7fc66e4652e/nilcm25m2021.tif; and 1 km
summary raster data: https://data-package.ceh.ac.uk/data/a3ff9411-3a7a-47e1-9b3e-79f21648237d.zip



## 7 Conclusion

The UK Land Cover Map series, comprising LCM1990 (formerly LCMGB) (Fuller *et al,*. 1994), LCM2000 (Fuller et al., 2002c), LCM2007 (Morton et al., 2011), LCM2017, LCM2018, LCM2019 and LCM2020 underpin a wide range of UK environmental science analysis and LCM2021 is expected to continue this trend. The accuracy of LCM2021 varies with class, but it has an overall accuracy of 82.6% for the 21 target classes and 86.5% for the 10 aggregate classes.

**Author contribution:** DM and CR acquired funding. CM, CR and DM pre-processed the data and conducted classifications. DM designed and implemented the Random Forest classification software and supporting computation structures with input from the whole team. CR, DM and CM developed code for pre-processing the satellite data. CM, CR and AO reviewed the classifications. CR, CM and AO prepared the validation data. CR and CM prepared the manuscript with contributions from all co-authors, DM, CR and CM designed the project. CM led the production of LCM2021.

## Competing interests

The authors declare that they have no conflict of interest.

## Acknowledgements

The authors would like to thank all the data collectors, processors, and providers who made this work possible, specifically: Satellite data provision: Contains modified Copernicus Sentinel data 2021.

Cartographic and DEM data for NI: Settlement development limits © Northern Ireland Statistics and Research Agency (NISRA) 2015. NI open data layers for Coastal water and Fresh water © Department of Agriculture, Environment and Rural Affairs, Northern Ireland. OSNI Digital Elevation Data and road network data Contains public sector information licensed under the terms of the Open Government Licence v3.0. Urban greenspace correction used greenspace areas for NI identified from OpenStreetMap data provided by OpenStreetMap and available under the Open Database License.

Cartographic data and DEM data for GB: Digital elevation data © Intermap Technologies Inc. or its suppliers 2003. OS open data layers - Contains OS data © Crown copyright and database right (2015). Boundaries from Rural Payments Agency © Crown copyright and database right and/or © third party licensors. Boundaries from Welsh Government, Department of Rural Affairs © Crown Copyright and database right and/or © third party licensors. Boundaries from Scottish Government © Crown Copyright and database right and/or © third party licensors. Contains OS Greenspace data © Crown Copyright [and database right] (2021).

Validation data: National Forest Inventory (NFI) Woodland GB 2018 data provided by the Forestry Commission, contains Forestry Commission information licensed under the Open Government License v3.0. Defra and the Rural Payments Agency are thanked for access to the IACS Agricultural data and Rural Payment Agency field survey data. We thank the Countryside



Survey team for provision of the Countryside Survey data. Countryside Survey data owned by UK Centre for Ecology &
Hydrology.
In addition, we thank Robbie Still of Kent Wildlife Trust for provision of the revised colour palette, and we thank all who
have been involved in the UK LCM since 1990, as this LCM builds on their hard work.

**Financial support**
This work was supported by the Natural Environment Research Council award number NE/R016429/1 as part of the UK-
SCAPE programme delivering National Capability.

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



# Appendix A: Notes on LCM2021 land cover classes

**Table A1: Discussion and commentary on each of the UK LCM2021 land cover and habitat classes. See Jackson (2000) for description of the underlying Broad Habitat classes.**

| LCM2021 land cover class | Notes |
| --- | --- |
| Broadleaved woodland | In the UK BAP *Broadleaved, mixed and yew woodland* broad habitat definition (Jackson, 2000) the broadleaved woodlands are characterised by stands >5 m high with tree cover >20%. Scrub (<5 m) requires a cover >30 % for inclusion. Such fine distinctions cannot be made through optical remote sensing. Open-canopy woodland (stands with trees <50 %) is a particular problem, albeit occurring relatively rarely in the UK; such areas are likely to be confused with other classes due to the dominance of the non-woodland vegetation and the sparsity of training areas representing these areas. <br><br> In the UK, broadleaved evergreen trees rarely occur in stands >0.5 hectares; an area large enough to create training areas suitable for classification. Consequently the classifier is likely to struggle with this land cover. These stands maybe classified as Coniferous woodland because of the full-year chlorophyll signal. <br><br> Mixed woodland stands of broad-leaved or evergreen trees exceeded the minimum mappable unit, they were treated as separate blocks within the woodland; in many parts of the UK, truly 'mixed woodlands' as opposed to those with mosaic-blocks of broadleaved and coniferous trees, are unusual. Stands with near-closed canopies can be interpreted easily in the field and pure examples can normally be found for training the classifier. |
| Coniferous woodland | The UK BAP *Coniferous woodland* class includes semi-natural stands and plantations, with cover >20 %. Classification of coniferous woodland is generally straightforward, but rare examples of open canopy semi-natural pinewoods are likely to be classified according to the dominant understorey class. <br><br> The UK BAP includes new plantation and recently felled areas. These are land use, not land cover. Newly felled areas are often dominated by grass, heather and encroaching vegetation and more likely to be classified as these, instead of coniferous woodland. Deciduous larch has potential for confusion with broadleaved deciduous woodland but is generally correctly identified. |
| Arable and horticulture | The BAP Broad Habitat *Arable and horticulture* includes annual crops, perennial crops such as berries and orchards and freshly ploughed land. This is a very broad class and as a consequence has large potential for spectral confusion with non-arable surfaces. The main confusion between arable and other classes occurs between arable land and improved grassland. This is especially likely when grassland is managed by cutting, followed by periods of low growth and reflectance from chlorophyll. When this happens the observed seasonal reflectance pattern can be similar to graminid crops, such as wheat and |





| | |
|---|---|
| | barley. Indeed grass managed in this way is technically a crop, so an arable classification is not necessarily wrong. |
| Improved grassland | Improved grassland is distinguished from semi-natural grasslands based on its higher productivity, lack of winter senescence, location and/or context. Grasslands lie on a continuum, so some confusion with other grassland types is inevitable. Confusion with grass-like crops will also occur. |
| Neutral grassland | The UK BAP Broad Habitat *Neutral grassland* is expected to be challenging for satellite-based classification. BAP *Neutral grassland* is defined by botanical composition and includes semi-improved grasslands managed for silage, hay or pasture (Jackson, 2000). There is not generally an obvious spectral difference between these and other productive grass types. However, the inclusion of Context Rasters for slope and distance to rivers appear to have helped greatly with Neutral Grassland detection. |
| Calcareous grassland | Calcareous grassland class is mapped spectrally. However, the inclusion of context layers for slope is expected to improve results. UKCEH does not have free access to a highly resolved soil PH/soil type layer, which we would expect to help further. For regions know to contain substantial coverage of Calcareous Grassland, for example Limestone Dales of Derbyshire and North Yorkshire, the South Downs and Salisbury Plain our results match expectations. |
| Acid grassland | The UK BAP *Acid grassland* can be spectrally variable, depending on dominant species composition. Deciduous acid grassland, dominated by *Molinea caerula* has a distinct signal from acid grasslands dominated by mixtures other grasses, rushes, mosses, herbs and sedges. In other work we have been able to refine this class successfully. However, we did not make this separation in historical maps, so we are not able to retrieve suitable observations from Bootstrap Training. <br><br> Bracken has a very distinctive spectral signal, but only at certain times of the year when its foliage begins to dominate its grassland understory. Historically, with restricted availability of satellite images we could not reliably separate the UK BAP *bracken* class from *acid grassland* so we combined these into a single land cover class. With the greater image frequency and therefore better access to seasonal signals it may now be possible to overcome this historic limitation, but to do this we will need novel training data as we will not be able to retrieve a signal from Bootstrap Training. |
| Heather; and heather grassland | For LCM2007 we refined the BAP *Dwarf shrub and heath* into two classes, depending on the density of heather, producing the heather and heather grassland classes (it is heather when there is greater than 25 % Heather Cover). This was to retain some consistency between the LCM1990 and LCM2000 classes open shrub heath and dense shrub heath. In some parts of the UK, significant areas of low-lying non-heather shrubs occur. For example, gorse can form a dominant shrub layer. |



| | |
|---|---|
| | Note: the land cover maps typically show confusion over heather, heather grassland and bog. However, they are often difficult to separate in the field. It is challenging to accurately estimate coverage above and below the defining threshold. |
| Fen, marsh and swamp | The UK BAP *Fen, marsh and swamp* includes fen, fen meadows, rush pasture, swamp, flushes and springs. From a remote sensing perspective fen, marsh and swamp is problematic as it is can be comprised of a wide range of vegetation types and many patches are below the MMU of the UKCEH land parcel spatial framework. The small size of many fen, marsh and swamp patches, plus their typically mosaic nature make it difficult to find reliable training data. Consequently, fen, marsh and swamp is likely to be underestimated in some regions. However, substantial areas of contiguous reed dominated fenland appear to be well detected. |
| Bog | The UK BAP *Bog* includes ericaceous, herbaceous and mossy swards in areas with a peat depth >0.5 m. We cannot detect peat depth from satellites. Vegetation on deep peat soils represent a continuum involving acid grassland, dwarf shrub heath and some types of fen, marsh and swamp and the separation of continuously varying land cover into discrete types can be difficult, especially when they exist in a complex small patch mosaic and their definitions are vague. |
| | We retain the bog class to maintain consistency with historical LCM products and the random forest classifier learns bog presence based on training data automatically generated from these. The predicted distribution occurs in regions where it is expected, so is a good indicator of where bog is likely to be occurring. However, bog and the range of upland vegetation classes expected to occur on peaty soils (acid grassland, fen marsh and swamp, heather, and heather grassland), potentially causing interclass confusion. This is partly due fine-scale variation but largely an effect of ambiguous definitions. UK BAP Broad Habitats (on which UKCEH land cover classes are based) were not defined with satellite remote sensing in mind. |
| Saltwater | Saltwater is rarely different spectrally from freshwater, and the saltwater distribution predicted by the random forest classifier is determined by coastal context rasters in Classification Scenes. There will be some confusion between saltwater and freshwater in tidal rivers, but not substantial. Occasionally, saltwater is confused with non-vegetated surfaces close to the coast and this happens because the automatically generated saltwater training classes coincide with the tide being out in the satellite view. The effect has so far been trivial but the result is that we predict saltwater with slightly lower accuracy than freshwater. Our main goal is to map land cover so coastal water and intertidal regions are not high priority. |
| Freshwater | The UKCEH Freshwater class comes from merging two BAP BHs (*Standing open water and canals,* and *Rivers and streams*) since they cannot be separated by spectra. In many cases, small and/or narrow water |





| | bodies fall below the MMU of the UKCEH land parcel spatial framework so effectively disappear into the dominant surrounding vegetation. Where these features are appropriately aligned and sufficiently wide, pixels they may be detected and if so will be available in the Raster Classification datasets. <br><br> Water bodies >0.5 ha and wider than 40 m are mapped with very high accuracy. The exceptions are temporary water bodies and quarries. Water in some quarries is strongly affected by the minerals in the rock and can result in atypical colours and misclassification. |
|---|---|
| Inland rock | The BAP Broad Habitat *Inland rock* covers both natural and artificial exposed rock surfaces which are >0.25 ha, such as inland cliffs, caves, screes and limestone pavements, as well as various forms of excavations and waste tips such as quarries and quarry waste. Opportunistic vegetation is common amongst rocky landscapes. We classify UKCEH inland rock if rock has the dominant signature. |
| Urban; and suburban | Within the *Built-up areas and gardens* BAP Broad Habitat we can reliably separate two UKCEH categories: urban and suburban. Urban includes dense urban, such as town and city centres, where there is little, if any, vegetation. Urban also includes areas such as dock sides, car parks and industrial estates. It is sometimes confused with other non-vegetated surfaces; for example open cast quarries or more rarely coastal rocks or ploughed fields. <br><br> Suburban includes suburban areas where the spectral signature is a mix of urban and vegetation signatures. suburban and urban lie on a continuum and confusion is expected. |
| Supra-littoral rock | Features that may be present in this coastal class include vertical rock, boulders, gullies, ledges and pools generally forming a narrow band when viewed from above. Only limited areas can be mapped using satellite remote sensing. |
| Supra-littoral sediment | This class includes sand dunes, which are reliably mapped. Areas of coastal sand may be confused between this class and the littoral sediment class. Supralittoral sediments can stabilise and from increasing volumes of vegetation. Heavily vegetated littoral sediment is likely to be classified as a vegetation class. |
| Littoral rock | These classes are those in the maritime zone on a rocky coastline. They are generally more extensive than supralittoral rock and thus more readily detected using satellite images. |
| Littoral sediment; and saltmarsh | The BAP Broad Habitat *Littoral sediment* has a subclass, the BAP Priority Habitat *Saltmarsh*. Saltmarsh is generally distinct from nearby vegetation and only occurs near the coast. As a consequence we can map this well with remote sensing. The saltmarsh class is occasionally subject to commission error, when we mistake other vegetation in the coastal zone (mainly Arable) as saltmarsh. <br><br> The littoral sediment is sometimes confused with the supra-littoral sediment class. |




## Appendix B: Display of LCM products

The UK Land Cover Map can be displayed however users require. However, standard and revised colour palettes are available (Tables B1 and B2) and are supplied as QGIS symbology files to enable users to rapidly display products.

**Table B1: Standard LCM colour palette.**

| Land cover class | Land cover class number | Red | Green | Blue |
|---|---|---|---|---|
| Broadleaved woodland | 1 | 255 | 0 | 0 |
| Coniferous woodland | 2 | 0 | 102 | 0 |
| Arable and horticulture | 3 | 115 | 38 | 0 |
| Improved grassland | 4 | 0 | 255 | 0 |
| Neutral grassland | 5 | 127 | 229 | 127 |
| Calcareous grassland | 6 | 112 | 168 | 0 |
| Acid grassland | 7 | 153 | 129 | 0 |
| Fen, marsh and swamp | 8 | 255 | 255 | 0 |
| Heather | 9 | 128 | 26 | 128 |
| Heather grassland | 10 | 230 | 140 | 166 |
| Bog | 11 | 0 | 128 | 115 |
| Inland rock | 12 | 210 | 210 | 255 |
| Saltwater | 13 | 0 | 0 | 128 |
| Freshwater | 14 | 0 | 0 | 255 |
| Supra-littoral rock | 15 | 204 | 179 | 0 |
| Supra-littoral sediment | 16 | 204 | 179 | 0 |
| Littoral rock | 17 | 255 | 255 | 128 |
| Littoral sediment | 18 | 255 | 255 | 128 |
| Saltmarsh | 19 | 128 | 128 | 255 |
| Urban | 20 | 0 | 0 | 0 |
| Suburban | 21 | 128 | 128 | 128 |

**Table B2: Revised colour palette avoiding use of red.**

| Land cover class | Land cover class number | Red | Green | Blue |
|---|---|---|---|---|
| Broadleaved woodland | 1 | 51 | 160 | 44 |
| Coniferous woodland | 2 | 0 | 80 | 0 |
| Arable and horticulture | 3 | 240 | 228 | 66 |
| Improved grassland | 4 | 1 | 255 | 124 |
| Neutral grassland | 5 | 220 | 153 | 9 |
| Calcareous grassland | 6 | 255 | 192 | 55 |
| Acid grassland | 7 | 178 | 145 | 0 |



| | | | | |
|---|---|---|---|---|
| Fen, marsh and swamp | 8 | 253 | 123 | 238 |
| Heather | 9 | 128 | 26 | 128 |
| Heather grassland | 10 | 230 | 140 | 166 |
| Bog | 11 | 205 | 59 | 181 |
| Inland rock | 12 | 210 | 210 | 255 |
| Saltwater | 13 | 0 | 0 | 92 |
| Freshwater | 14 | 0 | 0 | 255 |
| Supralittoral rock | 15 | 152 | 125 | 183 |
| Supralittoral sediment | 16 | 204 | 179 | 0 |
| Littoral rock | 17 | 255 | 255 | 128 |
| Littoral sediment | 18 | 255 | 255 | 128 |
| Saltmarsh | 19 | 128 | 128 | 255 |
| Urban | 20 | 0 | 0 | 0 |
| Suburban | 21 | 128 | 128 | 128 |

592

593





**Figure B1: Land Cover Map 2021 in revised colour palette (details of revised colour palette in Table B2).**