# Peer review of "LCM2021 – The UK Land Cover Map 2021"

_Earth System Science Data, 2023_

## Author Response (AR1)

**ESSD-2023-78: LCM2021 – The UK Land Cover Map 2021; Response to reviewer comments**

**Reviewer 1 comments:**

This paper is good summary of the state-of-the-art UK Land Cover product that the NERC has been producing since 1990. The methodology incorporates a wealth of contextual data to refine the scope of classification while using Google Earth Engine as the primary platform to process large amounts of data. The paper is well-presented and complete with easy access to the datasets through NERC's open data platform. The method is not novel in that it uses routine and traditional classifiers (random forest) and routine pre-processing steps (i.e. temporal compositing and only band reflectances were used). However, this is a signficant dataset that proves large-scale mapping at the national level can still achieve considerable accuracy using tried-and-tested methods. I see no significant gaps in their methodology. It can be accepted in its current form with just a very minor change in terminologies used for the Table heading below:

L233. Correspondence --> Confusion matrix. Correspondence is not the standard term for this table. Also font used is different. Change to Times new roman for the table
*Response: We thank the reviewer for their comments. Correspondence matrix has been changed to Confusion matrix. Text has been changed to Times New Roman.*

L242.Correspondence --> Confusion matrix. Correspondence is not the standard term for this table.
*Response: Correspondence matrix has been changed to Confusion matrix.*

**Reviewer 2 comments:**

The paper describes the process by which satellite data (Sentinel-2 L2A) and context data (DEM, coastline, and foreshore, tidal water, building, road, freshwater and forest layers) were combined to produce the LCM2021 map, including 10 m, 25 m and 1 km land cover products (21 classes, and a 10-class aggregated version). A Random Forest is trained to classify the pixels on a subset of selected polygons (the core training areas), and then used to produce the map. The results are validated on 32K high quality validation points.

The paper is well-written, with useful diagrams (e.g., the processing pipeline) and figures that help the presentation. Also, it includes detailed results of the classification process. However, I have a few major comments regarding the classification process.

1. In the manuscript, you stated that "LCM2021 methods replicate those for LCM2017 to LCM2020 with minor deviations to enhance accuracy.". According to the LCM2020 product description (https://www.ceh.ac.uk/sites/default/files/2021-11/lcm2020ProductDocumentation.pdf), the overall accuracy improved from 79.2% (LCM2020) to 86.5%. Apparently, some changes in the processing pipeline have been included (e.g., the purity across precedent years was set to 99%, instead of 80%). But, the model (Random Forest) has been used in both cases. Please, highlight the changes made and try to relate those changes to the performance improvement.

*Response: We thank the reviewer for their comments. Increases in classification accuracy have been made between LCM2020 and LCM2021, however the figures in the reviewer's comments are not directly comparable; the 79.2% accuracy figure for LCM2020 refers to the 21-class nomenclature, whereas the 86.5% accuracy figure for LCM2021 refers to the 10-class aggregate nomenclature. The accuracy for the LCM2021 21-class nomenclature is 82.6%, so whilst the classification accuracy for LCM2021 has been improved in comparison to LCM2020, the directly comparable level of increase is 3.4%, rather than 7.3%. We think these accuracy gains are due to improvements in the cloud-masking and training data distribution, particularly the increase in training areas in areas where the training data may have been too sparse previously, however we do not have quantitative evidence for this. The remit of this article is the LCM2021 data set and its production, not previous products, however the comment "LCM2021 methods replicate those for LCM2017 to LCM2020 with minor deviations to enhance accuracy." has been updated to "LCM2021 methods replicate those for LCM2017 to LCM2020, with minor deviations in cloud-masking processes and training data sourcing to enhance accuracy" to highlight these changes.*

2. You have trained a Random Forest to classify the land cover. However, state-of-the-art research papers often use deep learning-based approaches to do this task, showing a significant performance improvement.

*Response: This is an interesting suggestion. The strength of convolutional networks (as applied in references suggested by the reviewer) is primarily in understanding the structure of image patches. Urban areas having a high structural component, so such methods work well. However, many of the land cover classes we map are less structured and the information content is primarily in the spectral and temporal elements of the satellite data; Random Forest is highly suited and well-established for such tasks. However, work such as that by [1,2] suggest that 1-D convolutional networks may be more appropriate for our requirements, and we plan to test them in the future.*

*References:*

*[1] Pelletier, C., Webb, G.I. and Petitjean, F., 2019. Temporal convolutional neural network for the classification of satellite image time series. Remote Sensing, 11(5), p.523.*

*[2] Zhang, H.K., Roy, D.P. and Luo, D., 2023. Demonstration of large area land cover classification with a one dimensional convolutional neural network applied to single pixel temporal metric percentiles. Remote Sensing of Environment, 295, p.113653.*

3. The data set is imbalanced. Thus, please, include kappa to evaluate the accuracy performance.

*Response: Kappa has been added to Tables 3 and 4.*

And some minor comments:

a. It would be nice to have not only the majority, but also the top 5 predicted classes (with the respective probabilities), as this could help further evaluating machine learning models (e.g., use this data as a benchmark) and also to evaluate some other downstream machine learning tasks.

*Response: We thank the reviewer for their thoughts on our data and its potential reuse. In previous LCM versions (LCM2000 and LCM2007) the probability of the top 5 classes was included, however this information was scarcely used and so this aspect of the data set was dropped when LCM production moved to using Random Forests. Therefore the top five predicted classes and their probabilities are not given in LCM2021, however if there is demand from the user community for data of this nature, then we can look to include this in future LCMs.*

b. Would it be possible to colorize the confusion matrices (Table 3 and 4)?

*Response: We thank the reviewer for their suggestion and previous examples, however on this occasion we would prefer to keep the confusion matrix not colourized. We acknowledge that there are a variety of ways in which confusion matrices can be presented, however here we wish to preserve the absolute numbers of validation points within the confusion matrices, rather than normalising the values to a scale that would work better for colourisation, or presenting confusion matrices containing only producers accuracy as in the referee suggested reference Liu and Shi (2020). For the ranges in absolute values contained within the confusion matrix (0 to 10,102 for Table 4), colourisation would result in the majority of table cells comprising low values being presented as a similar colour, so not being particularly informative. Presenting confusion matrices un-colourized is also consistent with how confusion matrices have been presented in previous ESSD publications such as Zhang et al. (2021) (Tables 3-5), Li et al. (2023) (Table 3), and Liu et al. (2023) (Table 3).*

*References:*

*[3] Zhang, X., Liu, L., Chen, X., Gao, Y., Xie, S., and Mi, J.: GLC_FCS30: global land-cover product with fine classification system at 30 m using time-series Landsat imagery, Earth Syst. Sci. Data, 13, 2753–2776, https://doi.org/10.5194/essd-13-2753-2021, 2021.*

*[4] Li, B., Xu, X., Liu, X., Shi, Q., Zhuang, H., Cai, Y., and He, D.: An improved global land cover mapping in 2015 with 30 m resolution (GLC-2015) based on a multisource product-fusion approach, Earth Syst. Sci. Data, 15, 2347–2373, https://doi.org/10.5194/essd-15-2347-2023, 2023.*

*Liu, C., Xu, X., Feng, X., Cheng, X., Liu, C., and Huang, H.: CALC-2020: a new baseline land cover map at 10 m resolution for the circumpolar Arctic, Earth Syst. Sci. Data, 15, 133–153, https://doi.org/10.5194/essd-15-133-2023, 2023.*

---

## Author Response (AR2)

**ESSD-2023-78 Resubmission: 'LCM2021 – The UK Land Cover Map 2021' response to reviewer comments.**

*The authors thank the reviewers for their considered comments on our submitted manuscript. We are delighted that both reviewers are satisfied with the revisions that we have made to the manuscript, are that they are both now happy to see the manuscript accepted as is. The point-by-point comments below address the editor comments. Note that when author responses refer to specific lines in the manuscript, these refer to line numbers in the revised manuscript with tracked changes, rather than the revised manuscript where tracked changes have been accepted.*

**Topical editor decision: Publish subject to minor revisions (review by editor)**

Many thanks for the revision of your manuscript that fully satisfies the referees.
I have only a minor change request/ question to discuss with you before I will accept the article for final publication:
(1) the data availability statement still holds the information about the anonymous access to the data for referees. Is this intended to persist in the final version too? I would be fine with this, being a big supporter of open and freely-accessible data, but maybe CEH disagrees?

*Response: The intention is to remove the information referring to anonymous access in the data availability section (lines 313-321) from the final document as this referred to alternative access for data download specifically for the review process, whereby anonymous reviewer access to the data products was stipulated as a requirement. These anonymous links were time-limited for review purposes only, and will not be permanently available. However, the LCM raster products are, and will continue to be open and freely-accessible to the user community. The data is accessible via the UKCEH Environmental Information Data Centre (https://eidc.ac.uk/) with DOIs / links to the respective data download web pages contained within Table 6 in the submitted manuscript. To download the data via EIDC does, however, require a login. Consequently, lines 313-321 have been removed from the resubmitted manuscript. Lines 307-312 have also been removed as this text duplicated lines 297-302.*

(2) Would it be possible to add a sentence about the license of your data products? The DataCite metadata includes the link to all CEH licenses and it is quite difficult to decide under which of them your land cover files are published?

*Response: We have added the following sentence in section '6 Data availability' to clarify this: 'All LCM raster datasets are available under a single common licence without charge for non-commercial use which includes non-commercial research and use within public bodies and charities and their contractors. Alternative licensing can be arranged on request with commercial organisations who wish to use the datasets within their own internal business operations or to develop commercial products or services.'*

---

## Author Response (AR3)

UK Centre for Ecology & Hydrology
Lancaster Environment Centre
Library Avenue
Bailrigg
Lancaster
LA1 4AP
United Kingdom
Email: cmarston@ceh.ac.uk

5th September 2023

Dear ESSD,

The authors are delighted that our manuscript 'LCM2021 – The UK Land Cover Map 2021' (ESSD-2023-78) has been accepted for publication in Earth System Science Data. We would like to express our thanks to the anonymous reviewers for their constructive comments, and to the editor for consideration and acceptance of our article.

Yours sincerely,

Dr. Christopher Marston